# Non-Invasive Monitoring of Cutaneous Wound Healing in Non-Diabetic and Diabetic Model of Adult Zebrafish Using OCT Angiography

**DOI:** 10.3390/bioengineering10050538

**Published:** 2023-04-27

**Authors:** Jaeyoung Kim, Suhyun Kim, Woo June Choi

**Affiliations:** 1Research Institute for Skin Image, Korea University College of Medicine, Seoul 08308, Republic of Korea; jaykim830@gmail.com; 2Department of Dermatology and Skin Science, University of British Columbia, Vancouver, BC V6T 1Z1, Canada; 3Departments of Cancer Control Research and Integrative Oncology, British Columbia Cancer Agency, Vancouver, BC V5Z 1L3, Canada; 4Department of Biomedical Sciences, Korea University College of Medicine, Seoul 02841, Republic of Korea; dieslunae@naver.com; 5Zebrafish Translational Medical Research Center, Korea University Ansan Hospital, Ansan 15355, Republic of Korea; 6School of Electrical and Electronics Engineering, Chung-Ang University, Seoul 06974, Republic of Korea

**Keywords:** diabetic wound, wound healing, adult zebrafish, optical coherence tomography angiography

## Abstract

A diabetic wound presents a severe risk of infections and other complications because of its slow healing. Evaluating the pathophysiology during wound healing is imperative for wound care, necessitating a proper diabetic wound model and assay for monitoring. The adult zebrafish is a rapid and robust model for studying human cutaneous wound healing because of its fecundity and high similarities to human wound repair. OCTA as an assay can provide three-dimensional (3D) imaging of the tissue structure and vasculature in the epidermis, enabling monitoring of the pathophysiologic alterations in the zebrafish skin wound. We present a longitudinal study for assessing the cutaneous wound healing of the diabetic adult zebrafish model using OCTA, which is of importance for the diabetes research using the alternative animal models. We used non-diabetic (*n* = 9) and type 1 diabetes mellitus (DM) adult zebrafish models (*n* = 9). The full-thickness wound was generated on the fish skin, and the wound healing was monitored with OCTA for 15 days. The OCTA results demonstrated significant differences between diabetic and non-diabetic wound healing, involving delayed tissue remodeling and impaired angiogenesis for the diabetic wound, leading to slow wound recovery. The adult zebrafish model and OCTA technique may benefit long-term metabolic disease studies using zebrafish for drug development.

## 1. Introduction

Diabetes mellitus (DM), referred to as diabetes, is a chronic, progressive metabolic disorder that occurs when blood glucose levels increase because the body cannot produce sufficient amounts of the hormone insulin or cannot use the insulin effectively [1]. Diabetes is a significant public health problem for individuals from all countries. According to the International Diabetes Federation (IDF) Diabetes Atlas 10th edition report in 2021 [2], 537 million adults (20–79 years in age), or 1 in 10 of the global population, were estimated to suffer from diabetes worldwide, predicted to rise to 643 million by 2030 and 783 million by 2045. This disease was the direct cause of 6.7 million deaths among adults, accounting for more than 12% of deaths globally. Moreover, the global health expenditure on diabetes was estimated to be 966 billion dollars in 2021, having a significant economic impact on patients, their families, and national health budgets [2].

For patients with diabetes, all dermal wounds are a serious health concern and require careful attention. As a result of narrowed peripheral blood vessels or rarefaction of blood flow (peripheral vasculopathy), wound healing is impaired because less oxygen can reach the wound and the tissues do not heal as quickly as ordinary tissue. This slow wound healing increases the risk of bacterial or viral infections and other complications. Furthermore, high blood glucose (hyperglycemia) destroys nerves [3] making patients less sensitive to pain. With this loss of sensation, patients do not feel developing infections, which worsens wounds. Accordingly, even minor wounds caused by small cuts, burns, and insect bites can become serious for diabetic patients. If left untreated, diabetic wounds can develop into severe ulcers, leading to limb amputations [4]. Therefore, proper treatment and continuous management of diabetic wounds are needed to prevent complications.

Assessing wound healing is an essential step in the care of diabetic wounds. Wound healing is a dynamic and complex process that involves a highly coordinated cascade of events at overlapping phases, including hemostasis, inflammation, tissue formation, revascularization, and tissue remodeling [5,6,7]. For instance, when tissue is injured, the injury triggers an inflammatory reaction via cytokines deriving from platelet degranulation, fueling the immune-cell-driven inflammatory process in the injury site and stimulating the recruitment of fibroblasts. During the inflammatory phase and proliferative phase, the activated fibroblasts are differentiated into myofibroblasts, a type of highly contractile cells, responsible for producing abundant extracellular matrix (ECM) protein to facilitate wound closure as well as supporting the other cells associated with effective tissue remodeling [8,9,10,11]. However, this orderly sequence of the healing process is disrupted in chronic wounds, including those caused by diabetes. Thus, studies to better understand the pathology of impaired healing are beneficial to improving wound management.

Many wound healing studies have been performed in small animals such as mice with epidermal wounds because the animal models are amenable and effective in mirroring the pathological conditions observed in human wounds [12,13]. In recent years, researchers have increasingly used the zebrafish as an alternative to animal wound models [14,15]. The zebrafish (*Danio rerio*) is a popular research organism for studying human diseases because of its high degree of pathophysiological similarities to humans and unique features such as fecundity, easy care, and cost-effectiveness. Because of such advantages, various zebrafish models of diseases such as cancers [16], Parkinson’s disease [17], obesity [18], and diabetes [19] have been developed and harnessed to study human diseases. During wound healing, it exhibits all steps of adult mammalian wound repair [14] such that the adult zebrafish has recently been used as a model system for cutaneous wound healing research [20].

Living tissue imaging is helpful in evaluating the skin wound healing of the zebrafish because it enables observation of the dynamic range of cellular events directly (e.g., cutaneous angiogenesis) during wound repair. Fluorescence microscopy has been routinely used for imaging adult zebrafish wounds in vivo [14,20,21]. However, this assay requires site-specific fluorescence labeling or the use of transgenic reporter lines expressing fluorescence. Moreover, the imaging depth is superficial, so it is not easy to observe the healing of deep dermal wounds. Optical coherence tomography (OCT) is an interferometric imaging technology that detects light signals backscattered from the multi-layers within a live specimen to visualize the internal structures of the sample [22]. OCT can provide micrometer resolution, millimeter-deep tomographic images of the living tissues and organs using a label-free, non-invasive approach [23]. Because of its excellent reproducibility in measurement, OCT has been widely used to monitor the healing of skin wounds on small animal models [24,25]. Furthermore, OCT has recently been capable of imaging the blood vessels based on the motion contrast of circulating blood cells in the vessel lumen [26]. The advent of this functional OCT, called OCT angiography (OCTA) [27], has provided the opportunity to assess the additional vascular responses during wound healing, which is required for comprehensive evaluation [28,29,30].

In recent years, OCTA has been used to assess microvascular structure and function in diabetes of humans in vivo [31]. OCTA imaging has been performed on the feet and hands of healthy and diabetic patients, showing interesting microvascular complications of diabetes including neuropathy and microangiopathy as significant contributors to the development of chronic wounds and impaired wound healing [32,33]. However, OCTA study for wound healing of diabetic patients remains scarce due to difficulties in recruiting the cohort or wound management during healing, although the cutaneous wound healing of healthy humans has been reported using OCTA [28].

This study aims to demonstrate an OCTA to assess the diabetic wound healing of adult zebrafish. The full-thickness wound in the epidermis of the well-established zebrafish model of diabetes is longitudinally imaged using OCTA. Its tissue’s anatomical and vascular changes in wound healing are compared with those in non-diabetic wounds.

## 2. Materials and Methods

### 2.1. Zebrafish and Wounding

The following transgenic adult zebrafish (10–12 months old) was used in this study: Tg(ins:NTR:mcherry) [34], which expresses nitroreductase (NTR) in pancreatic beta(β)-cells. NTR is genetically fused to the red fluorescent reporter mCherry, allowing for visualization of the β-cells expressing NTR with fluorescence microscopy. The zebrafish were maintained at 28.5 °C with 14 h light and 10 h dark cycles. The zebrafish were anesthetized in 0.04% (0.04 mg/100 mL) tricaine (Sigma-Aldrich, St. Louis, MO, USA), and 2–4 scales in the trunk region above the anus were removed using forceps to generate wounds. The wound was generated using a 1 mm biopsy punch (#BP-10F, Kai Medical, Tokyo, Japan), and wounded fish were transferred and raised in cages during the wound healing experiment. All experimental procedures were approved by the Korea University Institutional Animal Care and Use Committee (KOREA-2021-0132) and were performed in accordance with the animal experiment guidelines of the Korea National Veterinary Research and Quarantine Service.

### 2.2. MTZ Treatment

For developing an adult zebrafish model of type 1 DM (T1DM), metronidazole (MTZ, Cat #M1547, Sigma-Aldrich) was dissolved in the embryo medium to yield a final concentration of 10 mM. A total of 20 μL of 10 mM MTZ solution was then intraperitoneally injected into the zebrafish; NTR converts MTZ into a cytotoxic compound, rapidly inducing apoptosis of NTR-expressing β-cells in the pancreas [35]. The destruction of insulin-producing pancreatic β-cells leads to hyperglycemia, a routine episode in T1DM [36].

### 2.3. Measurement of Blood Glucose Level

Individual zebrafish were anesthetized in 4% tricaine, and blood was collected without sacrifice using a heparinized glass needle along the body axis and anterior to the wound site in the region of the dorsal aorta. For blood glucose measurement, 2–3 μL of blood from a single zebrafish was collected on a clean parafilm. Blood glucose levels were measured using a commercial handheld glucometer (Gluconavii Link 0.3, SD Biosensor Inc., Suwon, Republic of Korea).

### 2.4. Fluorescence Imaging

Fluorescence imaging of the transgenic zebrafish was conducted to identify the ablation of β-cells in the pancreas. The pancreases of zebrafish were fixed overnight in 4% paraformaldehyde. For histological analysis, the fixed samples were embedded in 1.5% agar blocks containing 5% sucrose. Frozen blocks were sectioned into 10 μm thick slices using a cryostat microtome (Leica, Wetzlar, Germany), and the sectioned slices were rinsed with phosphate-buffered saline (PBS). Sectioned slices were stained with 4′,6-diamidino-2-phenylindole (DAPI, D1306, Thermo Fisher Scientific, Waltham, MA, USA) to counterstain the cells. All fluorescence images were captured using an A1Si laser-scanning confocal microscope (Nikon, Tokyo, Japan). Confocal images (1 μm z-stacking) were processed using NIS-Elements AR Analysis 4.30 software (Nikon).

### 2.5. OCTA Imaging

For cutaneous wound healing monitoring, this study was conducted using a commercial 1300 nm swept-source OCT (SS-OCT) system (VEG220C1, Thorlabs, Newton, NJ, USA) with a standalone probe head equipped with a pair of galvanometer scanners and a 10× telecentric scan lens (LSM02, 18 mm focal length, 3.4 mm working distance, Thorlabs, Newton, NJ, USA). A microelectromechanical system (MEMS) tunable vertical-cavity surface-emitting laser (VCSEL) was used as a light source emitting narrow-linewidth laser beams tuned across a 1300 nm centered broad spectral range (70 nm at 3 dB) at a sweep rate of 200 kHz (equivalent to an A-scan line rate), providing an axial resolution of 10.6 μm in tissue. The system sensitivity was ~98 dB at the sample surface with an incident light power of 12 mW. The lateral resolution was 7 μm around the depth of field (70 μm), and the imaging depth was ~1 mm for the skin tissue.

For OCTA imaging, the anesthetized zebrafish was carefully placed onto a plastic petri dish mounted on an adjustable titling stage under the OCT probe head (Figure 1a), where the zebrafish was slightly oblique to avoid strong specular reflection from their scales. The OCT probing beam spot was positioned at ~200 μm below the skin surface and then a small area (3.0 × 3.2 mm^2^) in the left flank of the zebrafish (red box in Figure 1b) that would involve the wound was imaged using a repetition scan protocol. Prior to the measurement, a preview OCT imaging was performed on the area to be imaged for 1 min. For the scan protocol, a B-scan consisting of 500 A-lines was repeated eight times at each of the 500 spatial locations along the elevational scan direction (y-axis), yielding an OCT data cube (1024 pixels (depth) × 500 A-lines (x-direction) × 4000 B-scans (y-direction)) through the scanned area, requiring an acquisition time of 42 s.

To generate a blood vessel map from the acquired data, we used an eigen-decomposition (ED)-based optical microangiography (OMAG) [37], one of OCTA algorithms dedicated to contrast tissue blood perfusion. The ED approach is a statistical analysis method utilizing the properties of spatiotemporal frequency components of the dynamic blood flow and surrounding stationary tissues from the time-varying complex OCT signals. The application of the method to the OCT data yielded three-dimensional (3D) datasets of tissue structure and corresponding vascular structure of the zebrafish. However, an en face (x-y) angiogram was generated to facilitate observation of the cutaneous vessel network by collapsing the 3D vasculature into 2D with maximum intensity projection (MIP).

Figure 1c,d illustrate representative OCT and OCTA cross-sections of the unwounded skin in the wild-type zebrafish, taken along the white line in Figure 1b, revealing typical anatomy and angio-structures within the trunk skin of adult zebrafish. The MIPs of OCT slabs with thickness of 50 μm, taken at different depths (green and red bars in Figure 1c) could be used to visualize the depth-resolved en face views of tissue layers, such as superficial scales (Figure 1e) and underlying pigmented stripe patterns, including melanophores (Figure 1f), as depicted in Figure 1b. Note that the thickness was estimated considering the refractive index of biological tissues on average (1.37) and the pixel size in the image (2.7 μm/pixel). Furthermore, the maximal projection of the OCTA dataset could delineate cutaneous blood vessels in the trunk of adult zebrafish, as depicted in Figure 1g. From the merged image (Figure 1h) of Figure 1e,g, the capillaries branching off arterioles and venules are located at the center of each scale (white dotted line), similar to previous results. Non-diabetic (control) and diabetic zebrafish wound sites were sequentially imaged at 3 h, 1 day, 4 days, 7 days, and 15 days post-injury using OCT/OCTA imaging.

### 2.6. Quantification of Vascular Response in the Wound Site

Vessel area density (VAD) of the wound site was measured as a vascular metric during wound healing to quantify vascular changes in the wound. We adopted a well-established vessel quantification algorithm often used to assess vascular response to skin wound healing [28]. Prior to quantification, a tissue segmentation technique was applied to the OCT data cube to extract a vascular slab at a specified depth [38]. The method relies on boundary detection of the first layer (i.e., skin surface) in the OCT image, from which the pixel locations of the tissue surface are detected by sliding a moving window (window size: 60 pixels) along the OCT A-line starting from the top pixel. Once the pixel locations of the surface boundary are known, a vascular slab at a certain depth can be segmented from corresponding OCTA cube, where the upper layer of the slab is determined by adding offset pixels from the surface and the lower layer is determined by adding the slab thickness to the upper layer. A resulting vascular slab was automatically extracted at depth of 200 μm below the skin surface and its thickness was 400 μm including hypodermis and muscle, where angiogenesis mostly occurred. Quantification was then performed on an MIP image of the vascular slab. A 2 mm diameter circular positive mask was applied to the MIP image, preserving the angiogenic capillary sprouts around or into the wound area while suppressing the other surrounding vessels, followed by binarization. Capillary VAD is defined as a unitless ratio of the total image area occupied by the vasculature to the total image area in the binary vessel map. The VAD of a small block with a 25 × 25 (pixels) kernel size was calculated to produce a spatially resolved VAD map, and the window kernel was then shifted across the entire binary image. The generated VAD map was resized to its original size and combined with the original binary image after Gaussian filtering with 3 × 3 (pixels) kernels.

### 2.7. Statistical Analysis

Data were averaged and represented as the mean ± standard error of the mean (SEM). The groups were compared using two-way ANOVA followed by Bonferroni’s correction for the multiple comparison. The *p*-value’s significance was set at less than 0.05. The graphs indicate statistical significance at three levels: * *p* ≤ 0.03; ** *p* ≤ 0.003; *** *p* ≤ 0.001.

## 3. Results

Full-thickness skin wound healing was studied in the wild-type (control) and diabetic adult zebrafish. We introduced wounds ~1 mm in diameter onto the flank of zebrafish using a dermatology punch with a diameter of 1 mm. Eighteen zebrafish (nine diabetic and nine non-diabetic) were prepared but the two diabetic zebrafish with low blood glucose levels were excluded, and thus, a total of sixteen zebrafish were included in the experiment. However, one diabetic zebrafish died during manipulation, so data were collected from six zebrafish after wounding in the diabetic model.

### 3.1. Identification of Diabetic Zebrafish Model

The T1DM model of Tg(ins:NTR:mcherry) transgenic zebrafish using MTZ was identified using confocal fluorescence imaging. Figure 2a illustrates the fluorescence images of control (left) and MTZ-treated zebrafish (right), revealing that, unlike the control (no MTZ), β-cells (red) were not visible in the MTZ-induced T1DM fish (+MTZ). This loss of β-cells in the pancreas is characteristic of T1DM, resulting in hyperglycemia (high blood glucose). The hyperglycemia was also checked by measuring the blood glucose levels, which were significantly higher in the diabetic model than in the control for time points of 4 dpi and 8 dpi, despite the level gradually recovering to normal within 14 days (Figure 2b) caused by β-cell regeneration by MTZ washout [34,39]. The measurements confirmed that the developed ablation model is appropriate for studying the short-term (~2 weeks) effect of primary diabetes.

### 3.2. Structural Changes during Cutaneous Wound Healing

OCTA imaging of the wounded normal fish (control, *n* = 9) and wounded diabetic fish (NTR/MTZ, *n* = 6) was conducted to assess structural and vascular changes in the wound during the wound healing period (15 days). Figure 3 illustrates microphotographs of the time-course of the self-wound healing of the control (up) and T1DM (down) fishes with full-thickness wounds on their flanks (red boxes), where the partial recovery (dotted circle) of stripe patterns in the wound site was seen at 15 days post-wounding (15 dpw) for the control, earlier than for the diabetic fish (Appendix A). Based on the superficial views, wound healing is likely slower in the diabetic model group than in the control group.

However, OCT can visualize microstructural changes in the wounded skin while being healed. Figure 4a illustrates the en face (x-y) OCT images of the wounded areas (red boxes in Figure 3), reconstructed by applying MIP to the OCT data cube of fish in each group. Figure 4b illustrates the cross-sectional (x-z) OCT images taken at the middle (white lines) of the wounds in Figure 4a, in which each of the structural features in the cutaneous wound regions of the control (Figure 4(b2–b7)) and the diabetic model (Figure 4(b9–b14)) were observed during the healing process. Initially, both groups lost all scale, epidermal, and dermal cells and part of the muscle tissue through the incision. Three hours post-wounding (3 hpw), a thin neo-epidermis (asterisks in Figure 4(b2,b9)) was created and covered the wound in both the control and diabetic models. The immediate re-epithelization and then wound closure precedes all other cellular responses during the healing process, in contrast to mammalian cutaneous wound healing, where the re-epithelization occurs at a relatively late phase of the repair process (i.e., following the inflammatory response and granulation tissue formation) [15,20]. At 1 dpw, a number of small bright spots (arrows) were found in the closed wound site (Figure 4(b3)) and we observed slow movements of the spots from the time-course B-scans repeatedly acquired at the same location. The micron-sized spots in motion may be floating particles such as, possibly, exudative inflammation cells such as neutrophils infiltrated into the wound for killing pathogens and subsequent phagocytosis [40,41]. During the subsequent days, the inflammatory reaction diminished, and the number of inflammatory cells slowly dropped for tissue regeneration [14]. From the fourth day (4 dpw), we observed that a granulation tissue (gt) appeared as a fine-grained region below the surface of the wound in each of the normal (Figure 4(b4)) and diabetic (Figure 4(b10)) fish. The appearance of the granulation tissue in the OCT image is quite similar to the histology of different fish skin wounds [42]. The granulation tissue was formed by accumulating fibroblasts migrating from the surrounding tissue into the wound space underneath the neo-epidermis, coinciding with collagen deposition and angiogenic sprouting (neovascularization). In normal adult zebrafish, tissue formation usually reaches the maximal size at 4 dpw and later regresses for tissue regeneration [5]. The OCT images of the control fish revealed that, during the healing progress, the granule-like tissue was progressively reduced (Figure 4(b4–b6)) and filled with new dermal and subdermal tissue resembling that of an unwounded region (m) on the 15th day (Figure 4(b7)). However, in the diabetic group, the granulation tissue (gt) remained until the 15th day, although it was downsized. Based on the OCT results, the tissue injury tends to heal slowly for the zebrafishes suffering from diabetes, affecting routine wound healing.

Revascularization (development of new blood vessels) in wound healing is a crucial response to cutaneous wounds, restoring an oxygenated blood supply and providing nutrients necessary for tissue regeneration. OCTA can reveal the neo-angiogenesis in the wound, as depicted in Figure 5. Figure 5(a1–a14) are the cross-sectional OCTA images corresponding to the cross-sectional OCT images (Figure 4(b1–b14)), delineating the blood vessels in perfusion. At 3 hpw, no blood vessels were seen in the wound and surrounding tissues for both fish groups (Figure 5(a2,a5)). This may be because of significant vessel loss in the full-thickness wound and the lack of perfused vessels around the damaged tissue. The vessels are still not visible at the inflammation stage (Figure 5(a3,a10)), and at 4 dpw, new microvessels (arrows in Figure 5(a4)) were concentrated around the granulation tissue (gt) in the control fish. Later, more vessels were distributed throughout the wounded area (Figure 5(a5–a7)), indicating revascularization in the wound at the proliferation phase [43]. However, for the diabetic model group, angiogenic vessels (arrows in Figure 5(a10)) were observed after the fourth day, but its population was less than that in the control group (Figure 5(a11–a13)). The results demonstrate that wound revascularization in diabetic fish is slower than normal.

Figure 5b illustrates en face OCTA images (angiograms) of the pre-wound and the wound regions of control and diabetic fish during wound healing. The angiograms were reconstructed with the MIP of the OCTA data cube obtained at each time point. In the control group, on the fourth day (4 dpw), many capillary vessels (arrow heads) sprouted around, growing into the wound space (dotted areas) on the seventh day (7 dpw) and diminishing until 10 dpw. The formation of the microvascular bed (neo-angiogenesis) and its regression were observed between the proliferation and subsequent maturation phases in normal wound healing for adult mammals, including the adult zebrafish [21]. In contrast, the angiograms of the diabetic fish revealed that the angiogenic vessels were scarce in their wound sites on day 4, and no significant vascular growth was seen from days 7 to 15. This insufficient vessel formation and growth may be caused by the impairment in angiogenesis, lagging the wound healing [44,45].

Moreover, the distribution of the angiogenic vessels was quantitatively analyzed by calculating the vessel area densities (VADs) in the wounds on different days for the control (*n* = 9) and diabetic (*n* = 6) zebrafish models. This study’s homebuilt processing and analysis software used the Matlab^®^ program to measure the VAD of the wound site during wound healing and quantified vascular changes in the wound. The method adopted a well-established vessel quantification algorithm and a tissue segmentation technique to extract a vascular slab at a specified depth, which allowed for accurate and precise measurements of the VAD. As described in Section 2.6, the OCTA slab was extracted at the depth range (hypodermis + muscle) demarcated as green dotted lines in Figure 6a. Figure 6b,c are the MIP views of the extracted OCTA slabs from control and diabetic fish. The circular wound area (2 mm in diameter) in the projection view was cropped and binarized, as depicted in Figure 6d,e, with which the VAD was calculated as depicted in Figure 6f,g, respectively. According to the ANOVA analysis (Figure 6h), two groups exhibited a similar trend in VAD changes over time when the vessel density surged after 1 dpw, reaching the maximum at day 4 and then declining gradually. The VAD curve is consistent with the results reported in the previous cutaneous wound healing studies [21,46]. However, on the fourth to tenth days (with revascularization occurring), the VADs in the diabetic group remained much lower than those of the non-diabetic group by over 50%, with statistically significant differences (at least *p* < 0.003) between groups. The differences are related to the disrupted angiogenesis in the diabetic wound seen in Figure 5. On the 15th day, no significant difference in the VADs was observed between the two groups because the zebrafish diabetic models built using MTZ resumed the production of the pancreatic β-cell normally from days 10 to 15 after MTZ treatment [47].

## 4. Discussion

In this study, we conducted the cutaneous wound monitoring of diabetic adult (10–12 months in age) zebrafish using OCTA technology to investigate the effect of diabetes on skin wound repair. Because of the unique capabilities of OCTA [48], the tissue microstructure and microvasculature of the small fish could be visualized non-destructively, enabling longitudinal monitoring of anatomical and vascular changes during skin wound healing in individual fish. The 15-day-long OCTA measurements of the non-diabetic (control) and diabetic wounds demonstrated that the wound-healing process of the diabetic group lagged behind the control group because of the delayed tissue regeneration (Figure 4b) and immature angiogenesis in the skin wound area was simultaneously suggested post-wounding (Figure 5b). In specific, at the initial healing phase, both the control and diabetic groups showed a loss of scale, epidermal, dermal cells, and part of the muscle tissue through the incision. However, three hours post-wounding, a thin neo-epidermis was created at the hemostasis phase and covered both groups’ wounds. One day post-wounding, small bright spots were observed in the closed wound site at the inflammatory phase, which were possibly exudative inflammation cells such as neutrophils infiltrated into the wound for killing pathogens and subsequent phagocytosis. During the subsequent days, the inflammatory reaction diminished, and the number of inflammatory cells slowly dropped for tissue regeneration. From the fourth day, at the proliferative phase, granulation tissue appeared as a fine-grained region below the wound’s surface in both the control and diabetic fish, coinciding with collagen deposition and angiogenic sprouting. The appearance of granulation tissue in OCT images was similar to the histology of different fish skin wounds. In normal adult zebrafish, tissue formation usually reaches the maximal size at four dpw and later regresses for tissue regeneration. At the remodeling phase, the OCT images of the control fish revealed that, during the healing progress, the granule-like tissue was progressively reduced and filled with new dermal and subdermal tissue resembling an unwounded region on the 15th day. However, in the diabetic group, the granulation tissue remained until the 15th day, although it was downsized.

Several adult zebrafish studies using OCTA have been reported for the different tissue organs of fish in vivo. Bozic et al. [49] reported OCTA imaging of the adult (>3 months in age) zebrafish’s ocular tissue to develop a method to quantify the biometry of zebrafish retinal vasculatures. Then, Yang et al. [50] and Lichtenegger et al. [51] demonstrated OCTA imaging of wild-type healthy adult (1–9 months in age) zebrafishes using polarization-sensitive OCT (PS-OCT) systems to characterize the scatter, polarization, and blood perfusion in the zebrafish skin. They monitored the healing process of the normal zebrafish’s cutaneous wounds, confirming the significant elevation of capillary vessel density in the wound area at the revascularization phase [50]. However, no OCTA studies for the diseased fish, especially zebrafish in diabetes, have been reported yet. This work was the first adult zebrafish study to demonstrate the adverse effect of the diabetic disease on wound healing, although the effect has been well documented by the previous OCT/OCTA studies of the animal diabetes models [46,52,53].

In this study, OCTA images of all fishes pre- and post-wound pertained to the signals throughout the cutaneous tissues. The background signals may be induced by tissue bulk motion or the dynamics inside the tissue. The tissue bulk motion is linked to global movements of the sample body caused by the heart beating or breathing, yielding strong signal artifacts all over the tissue in the OCTA image, of which the strength can be comparable or greater than the vessel signals. On the other hand, the dynamics inside the tissue are due to subcellular motion driven by the cellular organelle’s activities in the tissue cells, also yielding the background tissue signals in OCTA images. The subcellular dynamics in the tissue are inherent for a living sample, and can be readily detected by the intensity- or complex-signal-based highly sensitive OCTA manners including the ED approach used in this study [54].

However, Figure 5(a2,a9) had relatively low background tissue signals compared to other OCTA images because of the low signal-to-noise ratio (SNR) in corresponding OCT images (Figure 4(b2,b9)). The low SNR issue can occur as the confocal gate (i.e., focal volume) and coherent gate (i.e., coherence volume) are not properly matched in the sample arm; the fish tissue defocused is interfered within the coherent volume, resulting in blurred OCT tomograms, leading to the low contrast in OCTA image. For the vascular region, low-contrast OCTA images or angiograms can hamper the blood vessel analysis such as vessel area density (VAD).

The impaired angiogenesis seen in diabetic wound healing (Figure 5b) is the result of vascular deficits, the fundamental contributing factor to diabetes-associated diseases [44,45]. Previous animal studies found that diabetic wounds exhibited significant perturbation in the expression of factors that affect vascular regrowth, maturation, and stability. The expression of vascular endothelial growth factor A (VEGF-A), stimulating the formation of blood vessels, was downregulated in diabetic wounds [45,55]. Because the VEGF-A is derived by pericytes (perivascular cells) on the vascular tube, regulating the formation of a stable vascular network [56,57], the decreased expression of the pro-angiogenic factor indicates less coverage or even depletion of the pericytes [45], causing the resulting angiogenic defects.

Angiogenesis is the process of new blood vessel formation from pre-existing vessels, which plays a crucial role in wound healing by supplying oxygen and nutrients to the wound site. However, angiogenesis is a complex process that involves various molecular and cellular events, which can be affected by several factors, such as the type of wound, the location, and the severity. A study by Gurtner et al. reported that the angiogenesis process was delayed in a diabetic wound healing model, attributed to impaired endothelial cell function, and decreased vascular endothelial growth factor (VEGF) expression [58]. Another reason could be the impaired recruitment and activation of endothelial progenitor cells (EPCs), which play a crucial role in angiogenesis. Studies have shown that EPCs are decreased in patients with diabetes, which could contribute to the delayed angiogenesis process in diabetic wounds [59].

T1DM models were developed in adult or juvenile zebrafish using genetic manipulations [39,60,61] to simulate human-insulin-dependent diabetes. In this study, we used a popular Tg(ins:NTR:mcherry) zebrafish model expressing a fusion protein composed of enhanced red fluorescence and NTR under the control of the Ins promoter [39]. The MTZ treatment of the transgenic zebrafish led to selective ablation of insulin-secreting pancreatic beta cells (Figure 2a), resulting in hyperglycemia, followed by diabetes. However, blood glucose gradually declined back to a normal level at 14 days post-treatment (Figure 2b); normoglycemia was rapidly attained within two weeks after MTZ injection, consistent with previous results [47]. This means that the adult zebrafish recovered spontaneously from hyperglycemia after the chemical treatment without the need for insulin injection because of rapid regeneration of the pancreatic beta cells, in contrast to human T1DM or rodent models of extensive beta cell loss that require insulin therapy for survival [47,62]. Therefore, the elevation of VAD in the diabetic wound from 7 dpw (Figure 6f) may have resulted from the return to normalcy in the beta cell function for the diabetic zebrafish model. However, the VAD value of the diabetic wound at 15 dpw was less than the value pre-wound, indicating that more time was needed to fully recover to the baseline level.

However, the OCT probing beam spot was situated at the subsurface (at a depth of 200 μm from the skin surface) to cover the hypodermis to the top of muscle within the DOF, where the capillary sprouts in the wound’s neovascularization were well-focused, compromising with the blurring of the deeper and superficial vessels outside the DOF. To overcome the DOF limitation, it is recommended to use an electrically tunable lens (ETL), providing fast and dynamic control of axial focusing of the probe beam, benefiting the expansion of the DOF by over 1000 μm [63].

In this study, we used only vessel area density (VAD) to evaluate the vascular changes in wound healing. In addition to VAD, various vascular metrics such as vessel length density, vessel diameter, vessel tortuosity, and vessel branch point density can be readily applied to the OCTA data, ensuring more quantitative and comprehensive analysis of the cutaneous wound vasculatures. Recently, Untracht et al. [64] developed a new OCTA data analysis tool that is a graphical user interface (GUI)-based open-source toolbox to automatically measure such vascular metrics on an OCT angiogram. Adoption of this freely available tool will be positively considered for the next zebrafish study to characterize the diabetic wounds. Furthermore, the granulation tissue (gt) in the wound bed was observed to be morphologically distinct to the adjacent muscles (Figure 4b). Thanks to its characteristic textures, the granulation tissue in the OCT image may be separated from the surrounding tissues using texture image analysis techniques [65] and the area of the separated region could be used as a metric to compare the wound repair between the control and diabetic groups.

## 5. Conclusions

We demonstrated the longitudinal monitoring of cutaneous wound healing in a transgenic adult zebrafish model of T1DM in vivo using OCTA. During two weeks of wound healing, OCTA imaging exhibited characteristic microstructural and microvascular alterations in full-thickness skin wounds, distinct from the healing process in mammals. Based on the OCTA images, healing in diabetic wounds lagged or was abnormal in its phases compared with non-diabetic wounds. Therefore, the zebrafish model and OCTA technique will be beneficial for assessing chronic diabetic wounds and the therapeutic efficacy of new drugs for wound treatment. Future research in this field could explore potential interventions to improve diabetic wound healing in zebrafish, such as administering pro-angiogenic factors or stem cell therapy. Additionally, developing new OCTA technology and image analysis techniques could enhance the accuracy and reliability of wound monitoring in diabetic zebrafish models.

## Figures and Tables

**Figure 1 bioengineering-10-00538-f001:**
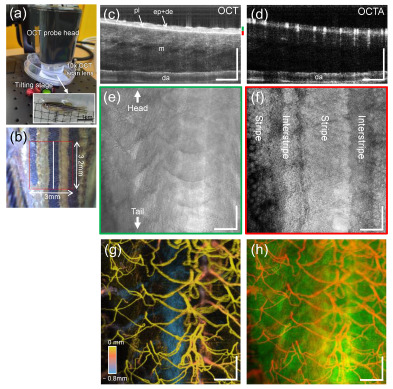
OCTA imaging of a transgenic adult zebrafish before wounding. (**a**) Anesthetized zebrafish placed on the tilting stage under the OCT probe head, and (**b**) closeup of a left flank of the zebrafish including a scanned region (red box, 3.0 mm × 3.2 mm). (**c**,**d**) Representative OCT and OCTA cross-sectional images obtained at a location (white line in (**b**)), exhibiting structural and vascular information (pl: pigment layer, ep + de: epidermis, m: muscle, da: dorsal aorta). (**e**,**f**) En face (x-y) views of the tissue layers segmented at two specific depths (green and red bars in (**c**)). (**g**) Pseudo-colored cutaneous vessel network of the zebrafish trunk obtained from the maximum intensity projection (MIP) of the OCTA dataset. The hotter color is closer to the surface. (**h**) Overlaid image of (**e**,**g**), where the white dotted line indicates a single scale involving the capillary network. Scale bars: 500 μm.

**Figure 2 bioengineering-10-00538-f002:**
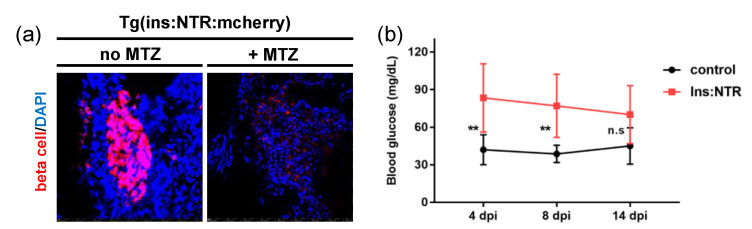
β-cell ablation and regeneration. (**a**) Fluorescence images of the control zebrafish beta cells compared with MTZ-treated zebrafish β-cells. (**b**) Blood glucose levels in the control and MTZ-treated zebrafish. β-cell ablation results that return to control levels by MTZ + 14 days. Significance levels are ** *p* < 0.003, and n.s *p* > 0.05.

**Figure 3 bioengineering-10-00538-f003:**
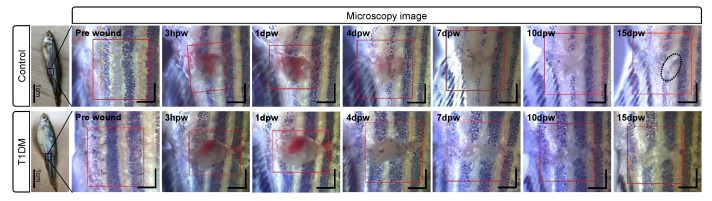
Bright-field microscopy images of the control (**up**) and T1DM (**down**) adult zebrafishes before wounding (pre-wound) and at 3 h, 1 day, 4 days, 7 days, 10 days, and 15 days post-wounding. Red boxes are the scanned areas involving the induced wounds. Scale bars: 1 mm.

**Figure 4 bioengineering-10-00538-f004:**
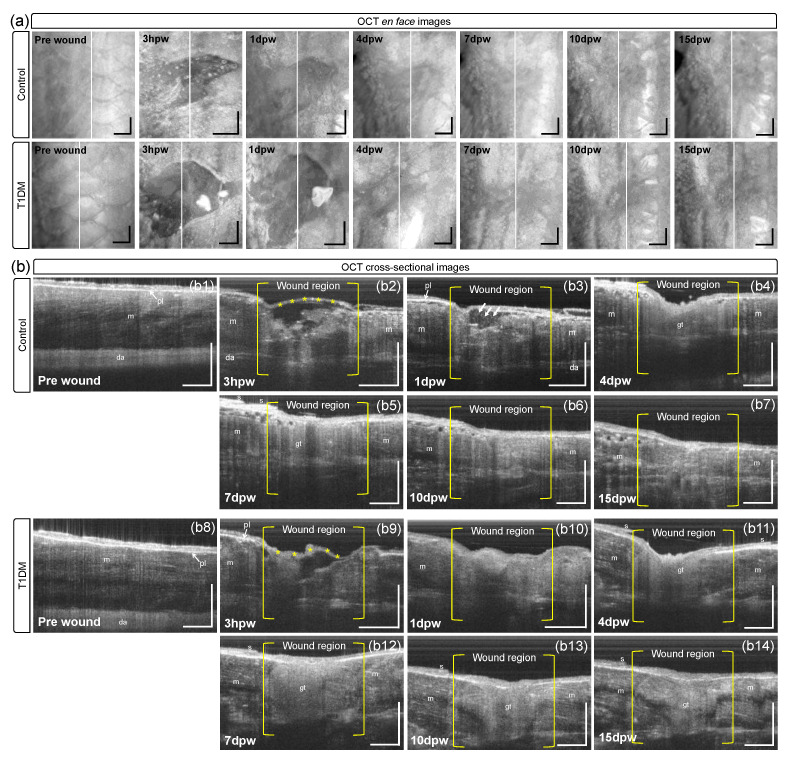
(**a**) En face OCT images of the red boxes in Figure 3. (**b**) Representative OCT cross-sectional images taken at white lines in (**a**), exhibiting structural changes in the wound regions of control (**b2**–**b7**) and diabetic fish (**b9**–**b14**) compared to their baselines (pre-wound), control (**b1**) and diabetic fish (**b8**). m: muscle, pl: pigmented layer, gt: granulation tissue, s: scale, da: dorsal aorta. Scale bars: 500 μm.

**Figure 5 bioengineering-10-00538-f005:**
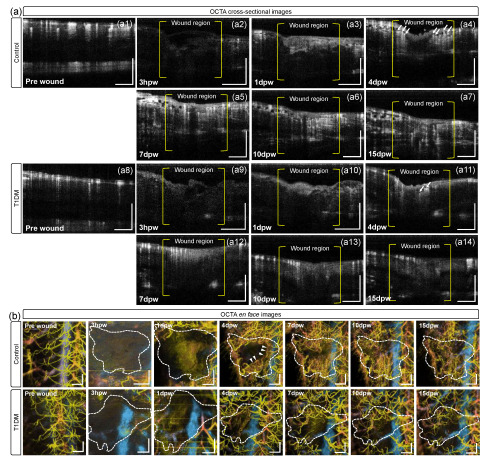
(**a**) Representative OCTA cross-sectional images obtained at the locations (white lines in Figure 2), corresponding to the OCT cross-sectional images in Figure 4b, exhibiting vascular information in pre-wound (**a1**) and post-wound regions of control (**a2**–**a7**), and pre-wound (**a8**) and post-wound regions of diabetic fish (**a9**–**a14**) over 15 dpw. (**b**) En face OCTA images (angiograms) of the time-course changes in the cutaneous microvascular networks in the wound regions (dotted areas) of control and diabetic fish over 15 dpw. The hotter color is closer to the skin surface. Scale bars: 500 μm.

**Figure 6 bioengineering-10-00538-f006:**
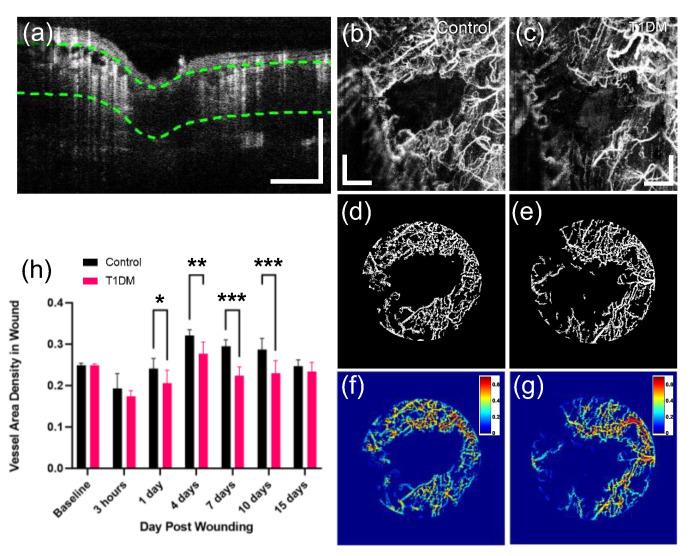
(**a**) Representative cross-sectional OCTA image of a 3D OCTA dataset. Based on the dataset, an OCTA slab is extracted at the depth range (green dotted lines), including the vascularized tissue under the epidermis. (**b**,**c**) MIP images of the extracted OCTA slabs in the control and diabetic zebrafish models. (**d**,**e**) Binary image of the wound area selectively cropped from (**b**,**c**). (**f**,**g**) Wound vessel area density (VAD) map calculated with (**d**,**e**). (**h**) The wound VADs (mean ± SEM) at different time points in non-diabetic (control) and diabetic groups. Significance levels are * *p* < 0.03, ** *p* < 0.003, and *** *p* < 0.001. Scale bars: 500 μm.

## Data Availability

Not applicable.

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
