# Peer review of "Non-Invasive Monitoring of Cutaneous Wound Healing in Non-Diabetic and Diabetic Model of Adult Zebrafish Using OCT Angiography"

_bioengineering, 2023, doi:10.3390/bioengineering10050538_

Round 1

Reviewer 1 Report

Dare really interestingear authors, excellent paper produced. 

The topic is really specific but all the outcomes described.

From the introduction to the methods extremely rigorous.

The only suggestion I make is to clarity in the manuscript the aim of the article. What is written in the abstract it's different from the introduction

This study is aimed to demonstrate an OCTA to assess the diabetic wound healing of adult zebrafish. The full thickness wound in the epidermis of the well-established zebrafish model of diabetes is longitudinally imaged using OCTA. Its tissue anatomical and vascular changes in wound healing are compared with those in non-diabetic wounds. 

Author Response

Once again, we are grateful that the reviewer acknowledged the improvements in the manuscript, and we thank the editor and the reviewer for their additional very constructive comments and criticisms. We have revised our manuscript as per the recommendations of the reviewers.

Point-by-point responses to your comments and changes made when revising the manuscript are provided below. The revised manuscript has been submitted to the website.

Here is a point-by-point response to the reviewer's comments and concerns.

We hope this modification meets your approval.

Reviewer 2 Report

Manuscript Number 2346801
"Non-invasive monitoring of cutaneous wound healing in non- diabetic and diabetic model of adult zebrafish using OCT angiography"

This study is well done and has representative results. The paper is well explained and has potential clinical applications if the OCT imaging can be used in patients with diabetic wounds to monitor wound progression and wound healing. However, the authors need to address to minor revisions that can improve the quality of the paper.

Please find my comments below:

Abstract

-        Authors have not mentioned what is already available in literature and how this study is different from others.

-        Mention the level of significance in the abstract

Introduction

-        Its known that fibroblasts are key players in wound healing. More information on role of fibroblasts in wound repair and regeneration and challenges in healing wound can be written from these papers:

1)     Hinz B. Formation and function of the Myofibroblast during tissue repair. J Invest Dermatol. 2007;127(3 Suppl):526-537.

2)     Monika P, Waiker PV, Chandraprabha MN, Rangarajan A, Murthy KNC. Myofibroblast progeny in wound biology and wound healing studies. Wound Rep Reg. 2021;1–17.

3)     Desmoulière A, Chaponnier C, Gabbiani G. Tissue repair, contraction, and the Myofibroblast. Wound Repair Regen. 2005;13(1 Suppl): 7-12.

4)     Monika, Prakash, Mathikere Naganna Chandraprabha, Annapoorni Rangarajan, P. Veena Waiker, and Kotamballi N. Chidambara Murthy. "Challenges in healing wound: Role of complementary and alternative medicine." Frontiers in Nutrition 8 (2022): 1198.

-        Are there any evidence of use of OCT imaging in humans for the current application (wound healing)?? If so, discuss in the manuscript.

Results & Discussion

-        It is suggested to include the data of neo-angiogenesis quantitatively (either in table or bar graph)

-        Was the wound closure determined qualitatively? i.e., by microscopic observation? What about quantification? % wound closure?? There are softwares that can measure the wound area (% wound closure). Authors are suggested to include that.

-        Authors can discuss little more elaborately on why angiogenesis (especially at cell and molecular level) is slow in wounded model referring to literature.

-        Discuss in detail regarding in which phase of wound healing process wound closure and angiogenesis was observed.

-        In conclusion it is T1DM and not T1D.

-        In conclusion ,mention the future scope of the work i.e., the work that needs to be carried out in near future which can lead to clinical implications & applications.

-        Check for spelling mistakes and grammatical errors throughout the manuscript.

Quality of English language looks fine. 

Author Response

(The authors gave the same response as above.)

Reviewer 3 Report

 In this work, it presented a longitudinal study for assessing the cutaneous wound healing of the diabetic adult zebrafish model using optical coherence tomography angiography. The adult zebrafish models were prepared. Full-thickness wound was generated on the fish skin, and the wound healing was monitored with OCTA for 15 days. The results demonstrated that the longitudinal monitoring of cutaneous wound healing in a transgenic adult zebrafish model of T1D in vivo using OCTA. The paper had a certain novelty, and its application of OCTA has a certain promotional value. This paper is likely of interest to readers focused on wound healing. In General, this work has a certain innovation. I think this paper could be accepted for publication in Bioengineering.

Author Response

We are writing to express my sincere gratitude for your time and effort in reviewing my paper. I was thrilled to receive your feedback and pleased to learn that no corrections were necessary. Your positive comments and constructive feedback have been invaluable, and I greatly appreciate your support.

Your expertise and insight have been invaluable to the success of my paper, and I am grateful for the attention you have given to my work. Your feedback has helped me to strengthen my research and writing skills and inspired me to continue to improve in my field.

Best Regards,

  1. Y. Kim, S. H. Kim, and W. J. Choi

Tel: +82-2-820-5901

Email: cecc78@cau.ac.kr
